# Personal protective equipment for reducing the risk of COVID-19 infection among healthcare workers involved in emergency trauma surgery during the pandemic: an umbrella review protocol

Dylan P Griswold [1,2] Andres Gempeler [3], Angelos G Kolias [1,2] Peter J Hutchinson [1,2] Andres M Rubiano [4,5]

For numbered affiliations see end of article.

**Correspondence to**
Professor Andres M Rubiano;
rubianoam@gmail.com

## ABSTRACT

**Introduction** Many healthcare facilities in low-income and middle-income countries are inadequately resourced and may lack optimal organisation and governance, especially concerning surgical health systems. COVID-19 has the potential to decimate these already strained surgical healthcare services unless health systems take stringent measures to protect healthcare workers (HCWs) from viral exposure and ensure the continuity of specialised care for patients. The objective of this broad evidence synthesis is to identify and summarise the available literature regarding the efficacy of different personal protective equipment (PPE) in reducing the risk of COVID-19 infection in health personnel caring for patients undergoing trauma surgery in low-resource environments.

**Methods** We will conduct several searches in the L·OVE (Living OVerview of Evidence) platform for COVID-19, a system that performs automated regular searches in PubMed, Embase, Cochrane Central Register of Controlled Trials and over 30 other sources. The search results will be presented according to the Preferred Reporting Items for Systematic Reviews and Meta-Analyses flow diagram. This review will preferentially consider systematic reviews of experimental and quasi-experimental studies, as well as individual studies of such designs, evaluating the effect of different PPE on the risk of COVID-19 infection in HCWs involved in emergency trauma surgery. Critical appraisal of eligible studies for methodological quality will be conducted. Data will be extracted using the standardised data extraction tool in Covidence. Studies will, when possible, be pooled in a statistical meta-analysis using JBI SUMARI. The Grading of Recommendations, Assessment, Development and Evaluation approach for grading the certainty of evidence will be followed and a summary of findings will be created.

**Ethics and dissemination** Ethical approval is not required for this review. The plan for dissemination is to publish review findings in a peer-reviewed journal and present findings at high-level conferences that engage the most pertinent stakeholders.

**PROSPERO registration number** CRD42020198267.

### Strengths and limitations of this study

► To the best of our knowledge this protocol provides a detailed description of the first systematic review on the effects of personal protective equipment in protecting emergency trauma surgery staff against COVID-19 infection.
► The protocol adheres to the Preferred Reporting Items for Systematic Reviews and Meta-Analyses Protocols guidelines for reporting a systematic review protocol.
► The protocol is being conducted by a multidisciplinary team with extensive experience in conducting high-quality systematic reviews.
► Given the rate at which new COVID-19-related studies are being published, there is the possibility that new studies will have been published at the time of review publication that were not available at the time of writing the review.

## INTRODUCTION

Many healthcare facilities in low-income and middle-income countries (LMICs) are inadequately resourced. COVID-19 has the potential to decimate these already strained surgical healthcare services unless health systems take stringent measures to protect healthcare workers (HCWs) from viral exposure. A recent study showed that 15.6% of confirmed patients with COVID-19 are symptomatic and that nearly half of patients with no symptoms at detection time will develop symptoms later.[1] Furthermore, the preoperative evaluation of emergency trauma patients is limited. These factors impede and confound diagnostic triage. Improper infection prevention may create a 'super-spreader' event in a high-volume healthcare facility or reduce available personnel. Consequently,

the infection control strategy of trauma surgery staff is a top priority.

To take care of patients, providers must first take care of themselves. Personal protective equipment (PPE) is paramount to protect HCWs from contracting the virus and becoming disease carriers. Basic recommended PPE for trauma surgery staff of high-income country facilities include (1) a surgical mask or better for all personnel interacting with patients and in the operating room (including cleaning staff); (2) N95 or better mask for all staff in close contact with the patients (<6 feet away); (3) powered air-purifying respirator for aerosolising and high-risk procedures (ear, nose, throat, thoracic and trans-sphenoidal neurosurgery operations); (4) universal testing of patients preoperatively to enable appropriate PPE use; and (5) changing scrubs after every procedure.[2] These recommendations are suitable for high-resource settings but are less feasible in low-resource settings. A rapid turnaround survey of 40 healthcare organisations across 15 LMICs revealed that 70% lack PPE and COVID-19 testing kits, and only 65% of the respondents showed confidence in hospital staff's knowledge about precautions to be taken to prevent COVID-19 infection among hospital personnel.[3]

Some resource-adjusted recommendations include the use of cloth masks and bandanas. While innovative, their moisture retention, reusability and filtration are considered very inferior to N95 and other masks.[4] What is most needed is evidence-guided recommendations for PPE use and COVID-19 screening in LMICs' surgical systems, where resources are either limited or unavailable. HCWs have been instructed to consider refraining from caring for patients in the absence of adequate PPE availability.

A preliminary search of PROSPERO (International Prospective Register of Systematic Reviews), MEDLINE, Cochrane Database of Systematic Reviews, and JBI Database of Systematic Reviews and Implementation Reports was conducted, and no current or underway systematic reviews on the topic were identified.

The primary objective of the review is to summarise the effects of different PPE in reducing the risk of COVID-19 infection among health personnel caring for patients undergoing trauma surgery. There is a need for high-quality evidence in this area, and a well-constructed systematic review can help provide a higher level of evidence.[5] Thus, the purpose of the review is to inform recommendations for the rational use of PPE in emergency surgery staff, particularly in low-resource environments where PPE shortages and high costs are expected to hamper the safety of HCWs and affect the care of trauma patients.

## METHODS
### Protocol registration
A protocol of this review following the Preferred Reporting Items for Systematic Reviews and Meta-Analyses (PRISMA) statement was registered in PROSPERO.

Any changes to the protocol will be amended in PROSPERO and reported in the final review. This review was conducted following the JBI methodology for systematic reviews.[6] The protocol adheres to the Preferred Reporting Items for Systematic Reviews and Meta-Analyses Protocols 2015.[7]

### Patient and public involvement
Patients and the public were not involved in the design of this systematic review protocol.

### Study design
A systematic review of peer review and grey literature following the PRISMA approach by Moher *et al*[8] is planned for this review. Figure 1 summarises the planned stages of the review as described in this protocol.

### Data source and search strategy
We will conduct several searches in the L·OVE (Living OVerview of Evidence) platform for COVID-19, a system that performs automated regular searches in PubMed, Embase, Cochrane Central Register of Controlled Trials and over 30 other sources. When compared with manual searches, this platform consistently identifies all the available studies associated with the terms of interest. It allows for a fast (automated) search that is easy to update—a crucial element given the urgent need to answer the research question rapidly and thoroughly. We will search for systematic reviews and randomised trials evaluating the effect of different PPE on the risk of COVID-19 infection in personnel involved in emergency trauma surgery during the pandemic. Other in-hospital clinical settings will be considered for inclusion and synthesis if evidence for trauma surgery setting is not available. Different clinical settings will be treated as subgroups from which extrapolation will be possible when considered adequate. Non-randomised studies will be considered if systematic reviews and randomised controlled trials (RCTs) are not available or are scarce and of low quality. We will include preprint studies identified in our searches, but no ongoing studies will be considered. Ongoing studies will be counted as excluded studies in the corresponding tables and PRISMA diagram. An example search for studies involving HCWs and N95 masks is provided in online supplemental appendix 1.

### Selection of studies
Following the search, all identified citations will be collated and uploaded into EndNote V.X9 (Clarivate Analytics, Pennsylvania, USA). The citations will then be imported into the JBI System for the Unified Management, Assessment and Review of Information (SUMARI) for review process. Two independent reviewers will examine titles and abstracts for eligibility. The full text of selected studies will be retrieved and assessed. Full-text studies that do not meet the inclusion criteria will be excluded, and a list of such excluded studies will be provided. Disagreements between the reviewers during title and abstract screening or full-text screening will be

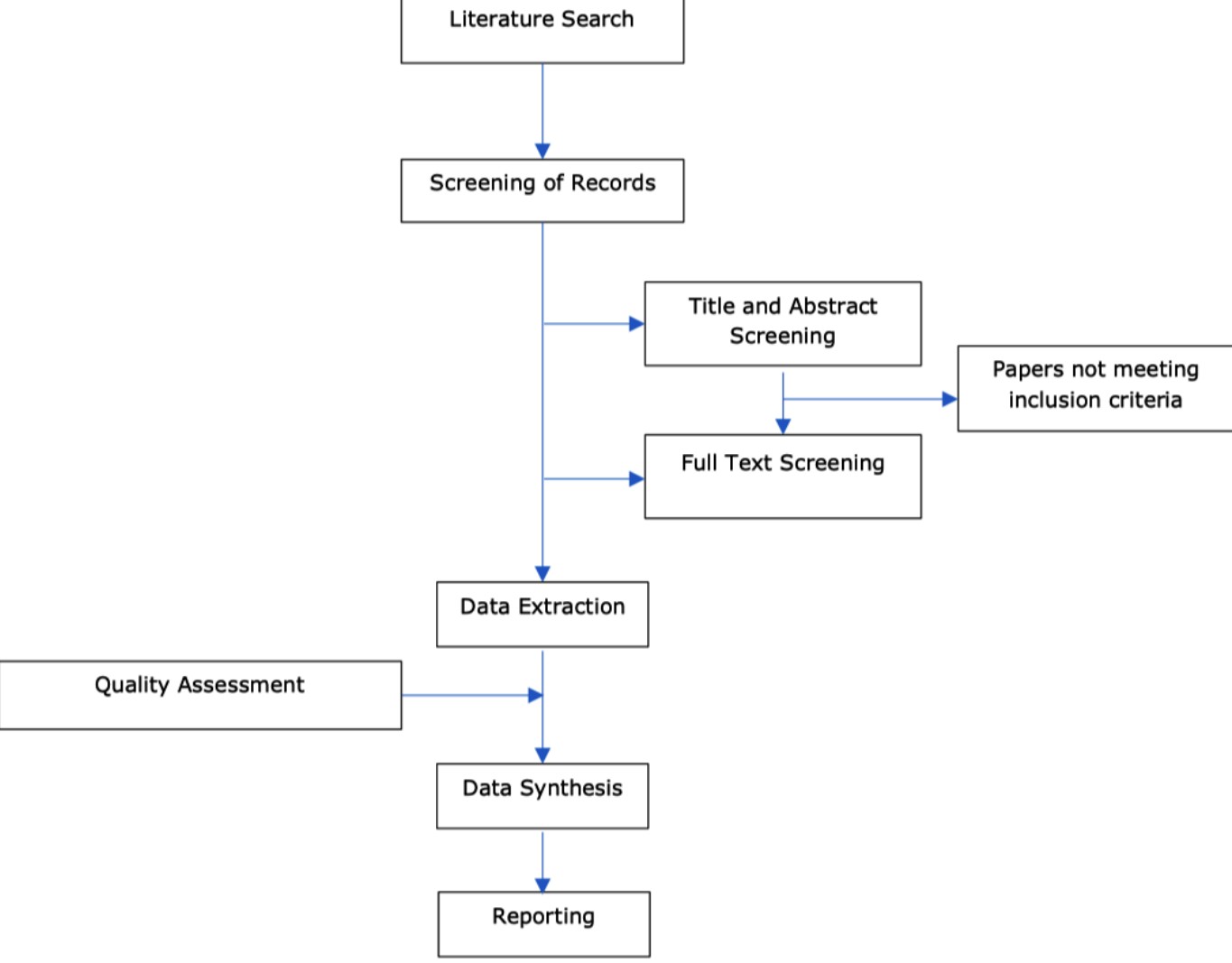

**Figure 1** Summary of search strategy.

resolved by consensus or with a third reviewer. The results of the search will be reported in full in the final report and presented in a PRISMA flow diagram.[8]

### Eligibility criteria/inclusion criteria

#### Participants

The review will preferentially include studies involving HCWs in emergency trauma surgery during the COVID-19 pandemic. Given the likelihood that reports on this specific population are scarce or even non-existent, if not available or insufficient we will consider studies of HCWs in any procedural and in-hospital setting such as emergency room and critical care management in COVID-19. Studies summarising the available evidence for other viral respiratory illnesses will be considered if COVID-19 evidence is not available and the setting reported is trauma surgery.

#### Intervention(s)

The intervention will be different types of PPE used when treating patients requiring emergency trauma surgery.

#### Comparator(s)

Comparators of interest will be no PPE use and different types of PPE.

#### Outcomes

This review will consider studies that include the following outcomes: risk of contagion to health personnel involved in the care of the described population during the COVID-19 pandemic; differences in surgical field vision; and expressed as incidence differences, risk ratios or ORs. All outcomes will be summarised narratively.

#### Types of studies

This review will consider systematic reviews of experimental and observational studies, and experimental or observational studies if not included in systematic reviews that fulfilled the population and intervention criteria. We will also include reports on implementation strategies that could inform recommendations for variable resource settings. Only studies published in English or Spanish will

**Table 1** PICOS inclusion criteria

| | |
|---|---|
| Participants | Healthcare workers in any procedural and in-hospital setting. |
| Intervention | PPE used by emergency trauma surgery staff. |
| Comparator | No PPE and different types of PPE within same class, that is, surgical masks vs N95. |
| Outcomes | Risk of contagion and surgical field vision. |
| Study type | Systematic reviews; experimental or observational studies not already included in the systematic reviews. |

PPE, personal protective equipment.

be included. We will include preprint studies identified in our search, but no ongoing studies will be considered.

The PICOS (participants, intervention, comparator, outcomes and study type) inclusion criteria is summarised in table 1.

## Quality assessment of included studies

Eligible studies will be critically appraised by one reviewer and verified with the other. We will use the AMSTAR tool to assess the risk of bias in systematic reviews, the Cochrane Risk of Bias-2 tool for RCTs and the ROBINS-I tool for non-randomised studies.[9–11] Risk of bias will be assessed only for the primary outcome: infection of HCWs. The results of the risk of bias assessment will be reported narratively and inform the grading of evidence summarised in the summary of findings (SoF) tables. Disagreements will be solved by consensus or by a third reviewer.

## Data extraction

Data will be extracted from the included studies by a reviewer and verified by a second reviewer using a data extraction tool from JBI SUMARI.[6] The data extracted will include specific details about the populations, study methods, interventions, and outcomes of significance to the review question and specific objectives. Disagreements will be solved by consensus.

## Data synthesis

Studies will be summarised narratively. Effect sizes from systematic reviews and from individual studies not included in them will be expressed as ORs (for dichotomous data) with their 95% CIs. Decision rules regarding data extraction for situations where data are:

▶ Reported at multiple time points: include all and note the time of measurement for each.
▶ Multiple 'doses': not applicable (all types of PPE will be recorded).
▶ Multiple exposures are compared (eg, ever exposed, frequency of exposure): as mentioned before, exposure will relate to the clinical setting of HCWs studied and will be classified accordingly.

For summarising results from other settings different from trauma surgery, the effect of PPE will be summarised by subgroups according to different clinical settings.

## Assessing certainty in the findings

The Grading of Recommendations, Assessment, Development and Evaluation (GRADE) approach for grading the certainty of evidence will be followed, and an SoF will be created using online software GRADEPro GDT 2020 (McMaster University, Ontario, Canada).[12 13] The SoF will present the following information where appropriate: absolute risks for the treatment and control, estimates of relative risk, and quality of the evidence based on the risk of bias, directness, heterogeneity, precision and risk of publication bias. The outcomes reported in the SoF will be risk of COVID-19 infection.

## Ethics and dissemination

No ethical approval will be required as this review is based on already published data and does not involve interaction with human subjects. The plan for dissemination, however, is to publish the findings of the review in a peer-reviewed journal and present findings at high-level international conferences that engage the most pertinent stakeholders. The proposed systematic review will provide a detailed summary of available evidence on the effects of different PPE in reducing the risk of COVID-19 infection of health personnel caring for patients undergoing trauma surgery. The purpose of the review is to inform recommendations for the rational use of PPE in emergency surgery staff, particularly in low-resource environments where PPE shortages and high costs are expected to hamper the safety of HCWs and affect the care of trauma patients.

## DISCUSSION

This protocol has been rigorously developed and designed specifically to assess the effects of different PPE in reducing the risk of COVID-19 infection among health personnel caring for patients undergoing trauma surgery. Given the limited recent evidence associated with the primary objective, findings from the review will be critical to researchers, policymakers, and government and non-governmental organisations in planning and developing guideline recommendations for PPE use in emergency trauma surgery settings, especially in LMICs. If protocol modifications are required, the authors will include the detailed description of any changes along with a justification during the publication of the review.

Clearly, in the era of COVID-19, where protecting HCWs from infection is essential, up-to-date information on the effects of PPE in protecting against COVID-19 infection is essential. This review will serve an important role as a repository of available evidence for the purpose of setting effective policy and clinical guideline recommendations.

**Author affiliations**
[1]NIHR Global Health Research Group on Neurotrauma, University of Cambridge, Cambridge, UK
[2]Division of Neurosurgery, Department of Clinical Neurosciences, Addenbrooke's Hospital, University of Cambridge, Cambridge, UK
[3]Clinical Research, Fundación Valle del Lili, Cali, Valle del Cauca, Colombia
[4]Neuroscience Institute, INUB-MEDITECH Research Group, El Bosque University, Bogotá, Colombia
[5]Neurological Surgery Service, Vallesalud Clinic, Cali, Colombia

**Contributors** AMR, PJH and AGK conceived the review. DG and AG designed the review. DG refined the review design. DG and AG were involved in the initial drafting of the manuscript. All authors were involved in subsequent draft manuscript reviews and updates and approved the final version of this protocol.

**Funding** This research was commissioned by the National Institute for Health Research (NIHR) Global Health Research Group on Neurotrauma (Project 16/137/105) using UK Aid from the UK Government. The views expressed in this publication are those of the author(s) and not necessarily those of the NIHR or the Department of Health and Social Care. PJH is the Chief Investigator of the RESCUEicp and RESCUE-ASDH randomised trials. DG was supported by the Gates Cambridge Trust. PJH was supported by a research professorship from the NIHR, the NIHR Cambridge Biomedical Research Centre, a European Union Seventh Framework Programme grant (CENTER-TBI; grant no 602,150), and the Royal College of Surgeons of England. AGK was supported by a Clinical Lectureship from the School of Clinical Medicine, University of Cambridge, and the Royal College of Surgeons of England. AG was supported by a grant from the Clinical Research Centre, Fundación Valle del Lili.

**Competing interests** None declared.

**Patient consent for publication** Not required.

**Provenance and peer review** Not commissioned; externally peer reviewed.

**ORCID iDs**
Dylan P Griswold http://orcid.org/0000-0003-0291-8360
Andres Gempeler http://orcid.org/0000-0001-9217-9500
Angelos G Kolias http://orcid.org/0000-0003-3992-0587
Peter J Hutchinson http://orcid.org/0000-0002-2796-1835
Andres M Rubiano http://orcid.org/0000-0001-8931-3254

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
