## [Reviewer comments · BMJ Open]

ARTICLE DETAILS

TITLE (PROVISIONAL)	Personal protective equipment for reducing the risk of COVID-19 infection among healthcare workers involved in emergency trauma surgery during the pandemic: an umbrella review protocol
AUTHORS	Griswold, Dylan; Gempeler, Andres; Koliass, Angelos; Hutchinson, Peter; Rubiano, Andres M

VERSION 1 – REVIEW

REVIEWER	Dr Adrian Heald University of Manchester and Salford Royal Hospital
REVIEW RETURNED	24-Nov-2020

GENERAL COMMENTS	Overall comments This is a very important area in which to evaluate the evidence as recently described in another paper: Tian Z, Stedman M et al. Personal protective equipment (PPE) and infection among healthcare workers - What is the evidence? Int J Clin Pract. 2020 Nov;74(11):e13617. doi: 10.1111/ijcp.13617. I would suggest that this relevant 2020 review is cited. The matter of resource availability in low and middle income countries is particularly pertinent in relation to the safety of health care professionals and transmission of the virus in health care settings. There has been a sense since March 2020 of 'every person for themselves' in international terms. This paper can contribute to some degree of rebalancing, in relation to appreciation that this is a global health issue. Emergency trauma management is a key area in relation to systematic review of the evidence base and will be addressed in the analysis proposed. Specific points that merit amendment: Introduction: Please define all abbreviations eg LMIC and PAPR. Discussion Some mention of the recent announcements re the C19 virus vaccine would be very relevant at this point and should be included in the paper.
---

REVIEWER	Davide Piaggio University of Warwick, United Kingdom
-----------------	---

REVIEW RETURNED	30-Nov-2020
-------------

GENERAL COMMENTS	A protocol for a very interesting and ambitious systematic literature review and metanalysis surrounding the efficacy and effectiveness of PPE to prevent COVID-19 in hospitals and surgical theatres. The results from this study could be used as evidence by policy makers, when drafting guidelines. A few comments that you may consider: Are the methods described sufficiently to allow the study to be repeated? I answered No, because the search string should be reported to facilitate the repetition of the study (if not in the protocol, please consider this for the final publication). You may also consider reporting your exclusion/inclusion criteria in a table. As regards the questions about the results and discussion, I assigned it a N/A, because it can only be reviewed when the review and metanalysis will be performed and published.
---

VERSION 1 – AUTHOR RESPONSE

Thank you for considering our manuscript, Personal protective equipment for reducing the risk of COVID-19 infection among healthcare workers involved in emergency trauma surgery during the pandemic: a systematic review protocol, for publication with BMJ Open. Given the urgent nature of the situation, our final review was recently accepted for publication with *Journal of Trauma and Acute Care Surgery*. While it is unusual for the protocol to be published after, or at the same time, of the review, we believe there is merit in continuing to publish our protocol. We spent a lot of time and effort creating a thorough protocol for our review and believe it serves as an exemplary template for creating a robust systematic umbrella review protocol, of which there are very few. By following it carefully, we were able to develop a high-level review with a higher-than-average certainty in the evidence. We were then able to publish it in a high-quality journal within our field, as directed in the protocol. There is also methodological guidance in the protocol that is not as detailed in the review. Taking the reviewers' suggestions, we created a PICOS table in addition to an Appendix outlining the process for conducting a search in the L-OVE COVID-19 platform. This aids researchers in reproducing our search strategy in addition to aiding researchers in carrying out other COVID-19 relevant reviews using the L-OVE COVID-19 platform.

The authors would like to thank Dr. Heald for his helpful comments. A point-by-point reply follows.

1. We took your advice and cited the paper by Tian et al. in the introduction, noting, "*There is a need for high-quality evidence in this area, and a well-constructed systematic review can help provide a higher level of evidence*" (Tian et al.).
2. Thank you for pointing out the missing context for the abbreviations. We have made the necessary changes.
3. We agree that it is important to discuss the C19 vaccine; however, given the time sensitive nature of the topic, our review has already been accepted for publication. Since we did not discuss the vaccine in the review, we believe it is prudent to leave it out of the protocol.

The authors would like to thank Dr. Piaggio for his helpful comments. A point-by-point reply follows.

1. We agreed that there is insufficient information provided so as to allow the repetition of the study. Given the time-sensitive nature of the topic and the rapid review process for COVID19

papers in most academic journals, our review has already been accepted for publication. To aid the study's reproducibility and future C19 studies using the LoVE platform, we added an Appendix with a walk-through search strategy that was not included in the review. Thank you for pointing this out.

2. We agree that a table outlining the inclusion criteria would be helpful. Thus, we summarized the inclusion criteria as a PICOS table.

Thank you, once again, for considering our protocol for publication in BMJ Open. Methodology is arguably the most critical aspect of conducting a review, and we believe our protocol serves as an exemplary systematic umbrella review template. Our protocol was the guidepost through which we drove the review. We hope other researchers find our protocol development helpful in designing their own protocols to write high-quality reviews of their own.